# Effect of Paternal Diet on Spermatogenesis and Offspring Health: Focus on Epigenetics and Interventions with Food Bioactive Compounds

**DOI:** 10.3390/nu14102150

**Published:** 2022-05-21

**Authors:** Gabriela de Freitas Laiber Pascoal, Marina Vilar Geraldi, Mário Roberto Maróstica, Thomas Prates Ong

**Affiliations:** 1Food Research Center (FoRC), Department of Food Science and Nutrition, School of Pharmaceutical Sciences, University of São Paulo, Sao Paulo 05508-000, Brazil; laiber@usp.br; 2Department of Food Science and Nutrition, School of Food Engineering, University of Campinas, Campinas 13083-862, Brazil; marinavilar35@gmail.com (M.V.G.); mmarosti@unicamp.br (M.R.M.J.)

**Keywords:** paternal nutrition, bioactive compounds, spermatogenesis, epigenetics, offspring

## Abstract

Infertility is a growing public health problem. Consumption of antioxidant bioactive food compounds (BFCs) that include micronutrients and non-nutrients has been highlighted as a potential strategy to protect against oxidative and inflammatory damage in the male reproductive system induced by obesity, alcohol, and toxicants and, thus, improve spermatogenesis and the fertility parameters. Paternal consumption of such dietary compounds could not only benefit the fathers but their offspring as well. Studies in the new field of paternal origins of health and disease show that paternal malnutrition can alter sperm epigenome, and this can alter fetal development and program an increased risk of metabolic diseases and breast cancer in adulthood. BFCs, such as ascorbic acid, α-tocopherol, polyunsaturated fatty acids, trace elements, carnitines, N-acetylcysteine, and coenzyme Q10, have been shown to improve male gametogenesis, modulate epigenetics of germ cells, and the epigenetic signature of the offspring, restoring offspring metabolic health induced by stressors during early life. This indicates that, from a father’s perspective, preconception is a valuable window of opportunity to start potential nutritional interventions with these BFCs to maximize sperm epigenetic integrity and promote adequate fetal growth and development, thus preventing chronic disease in adulthood.

## 1. Introduction

Infertility is defined by the World Health Organization (WHO) as the failure to achieve a pregnancy after at least 12 months of regular unprotected intercourse [1], and it has become a growing health concern for couples in present times. It is estimated that 20–70% of fertility problems are caused by the male partner, and at least 30 million men worldwide are infertile [2,3,4].

Men experience a decrease in fertility potential during aging [5]. Importantly, imbalanced nutrition, including excessive intake of calories and malnutrition (low intake of fibers, vitamins, and BFCs), together with the lack of physical activity, contributes to body fat accumulation and the development of non-communicable diseases, such as cardiovascular diseases, diabetes, and cancer [6]. Furthermore, lifestyle factors such as smoking, alcohol abuse, and poor nutritional intake can negatively impact the quality of sperm parameters, such as semen volume, sperm motility, and quality, promoting a decline in male fertility potential [7,8,9,10,11,12].

BFCs can be defined as nutrients and non-nutrients present in the food matrix that can produce physiological effects beyond their classical nutritional properties [13] and have emerged as a potential treatment for male infertility. Many vitamins, trace elements, and other BFCs obtained through the diet have been shown to participate in different processes involved in male reproductive function [14,15,16,17]. These BFCs are comprised of molecules with antioxidant properties that are involved in cell protection from damage due to reactive oxygen species (ROS), mediate inflammation, and support the antioxidant defense system, which can prevent reduced sperm motility, reduced sperm count, and abnormal morphology [18,19].

Recently, increased interest has been directed towards understanding epigenetic regulation in male reproductive physiology [20]. Epigenetics is, by definition, the gene regulation process without changes in DNA sequence and includes DNA methylation, posttranslational histone modifications, and non-coding RNAs, including microRNA (miRNA) regulation [21].

Epigenetic changes occur during spermatogenesis, including significant reorganization of sperm chromatin structure, thus allowing the sperm cell to become highly specialized [22]. Therefore, spermatogenesis is particularly vulnerable to epigenetic alterations. Pre-puberty/puberty and adulthood comprise developmental windows in which the epigenome would be especially plastic and susceptible to changes induced by environmental factors, such as the male diet [23,24].

BFCs, such as ascorbic acid, α-tocopherol, polyunsaturated fatty acids (PUFAs), trace elements, carnitines, N-acetylcysteine, coenzyme Q10, and folate have been evaluated to improve male gametogenesis, modulate epigenetics of germ cells, and the epigenetic signature of the offspring, restoring offspring metabolic health induced by stressors during early life. Thus, this review will focus on epigenetics and interventions with food bioactive compounds, in the paternal diet, on spermatogenesis and offspring health based on clinical and in vivo studies.

## 2. Diet and Male Reproductive Health

Spermatogenesis is a complex process that involves the continuous production of sperm cells in the seminiferous tubule [25]. After puberty, spermatogonial stem cells (SSCs) provide the foundation of sperm cells, a process that persists throughout the majority of a male’s lifetime [26]. A fertile man produces over 200 million sperm cells daily within the testis [25]. The testis is also an endocrine organ where high levels of testosterone are produced, supporting normal spermatogenesis and male phenotypic characteristics [27]. Spermatogenesis consists of four differentiation stages: 1 (mitotic): SSCs undergo mitotic proliferation, resulting in primary spermatocytes; 2 (meiotic): secondary spermatocytes undergo meiosis; 3 (post-meiotic): to form haploid spermatid cells; 4 (mature sperm): spermatids undergo spermiogenesis. Spermiogenesis is the final maturation stage for elongated sperm cell formation: a mature sperm cell capable of fertilization [27]. These stages involve many cellular events in the testis, which are regulated by hormonal and signaling pathways [22,28].

The steroid testosterone is the main hormone essential to maintain spermatogenesis, as it acts in the testis regulating spermatogenesis [29]. Testosterone is required for critical processes: maintenance of the blood-testis barrier, adherence of elongated spermatids to Sertoli cells, spermatocytes meiosis process, and the release of mature sperm cells [30]. The Leydig cells produce testosterone, which diffuses into the seminiferous tubules, peritubular myoid cells, Sertoli cells, and Leydig cells, as well as into the blood vessels. The Sertoli cells mediate metabolic factors and signals required for the proliferation and differentiation of germ cells [29,31]. Peritubular myoid cells also provide a basement membrane in the testis for SSCs that produce germ cells, which will develop into sperm cells [32].

### 2.1. Effects of BFCs on Male Spermatogenesis

Besides genetic background, nutritional and lifestyle factors play a key role in reproductive health and can influence fertility [33]. Adverse environmental factors in a man’s life, such as malnutrition, obesity, sedentary lifestyle, stress, alcohol intake, smoking and drug abuse, and exposure to pollution or radiation make the man more susceptible to developing reproductive pathological conditions, including subfertility or infertility [7,33,34,35].

It is well-known that consuming a diet rich in BFCs can influence men’s fertility [34]. Following a nutritious diet can help maintain a healthy weight and prevent obesity, a condition that is associated with increased oxidative stress (OS), the main causative factor of infertility [35]. OS is a condition associated with increased generation of ROS and reduced cellular antioxidant capacity. In seminal plasma, this condition is characterized by increased nuclear DNA damage and lipid peroxidation, as well as decreased levels of antioxidant enzymes such as catalase, glutathione peroxidase, and superoxide and vitamins A, C, and E [11]. Sperm cell plasma membranes are constituted of phospholipids and PUFAs, and their cytoplasm contain low concentrations of scavenging enzymes, which makes them particularly susceptible to oxygen-induced damage [18]. Excessive generation of ROS results in damage to the sperm cell plasma membrane and subsequent loss of sperm cell quality and function, such as the deregulation of capacitation, activation, motility, counts, and sperm-egg fusion [19].

Several BFCs, such as vitamins, minerals, fatty acids, amino acids, and antioxidants, have been shown to improve sperm cell physiology and function through multiple mechanisms, including reduction in ROS, restoration of the antioxidant defense system, and inhibition of inflammation. The BFCs that improve sperm cell quality and function are summarized in Table 1 with their reported outcomes.

#### 2.1.1. Vitamins

Vitamin A, or retinoids, comprise a group of natural antioxidants that inhibit lipid peroxidation and protect against cell damage [62]. They are necessary for a normal spermatogenic process by stimulating the transcription of genes involved in meiosis [63]. Men with normal sperm parameters presented greater retinol serum concentrations than those with low sperm cell count (oligozoospermia) and impaired motility (asthenozoospermia) [36].

Vitamin C (ascorbic acid) is found in fruits and vegetables and has the functionality to reduce DNA damage directly by scavenging free radicals and decreasing lipid peroxidation [64]. It comprises a key component in the antioxidant system of seminal plasma. According to Colagar et al. (2009), men with idiopathic infertility have lower levels of vitamin C in their seminal plasma than fertile men, which is a condition that can be a risk factor for infertility and abnormal sperm cell morphology [37]. Vitamin C supplementation (2 mg/day for 2 months) in infertile men improved sperm cell count, motility, and morphology, indicating that seminal vitamin C may improve infertility issues in infertile men [38].

Vitamin E is the generic term for a group of tocopherols and tocotrienols found in vegetable oils. It is an important antioxidant component and a major cell protector of the integrity of PUFAs in the cell’s membrane against ROS, thus maintaining their bioactivity [65]. Lower serum α-tocopherol levels have been found in men with oligozoospermia and asthenozoospermia [36]. Matorras et al. (2020) showed that vitamin E supplementation (400 mg α-tocopherol/day) for three months was positively associated with live birth rate and a trend towards better results in in vitro fertilization parameters, but it did not significantly increase progressive sperm cell motility in a double-blind, placebo-controlled, randomized study [39]. In another clinical trial with infertile men, Keskes-Ammar et al. (2003) demonstrated that daily supplementation with vitamin E (400 mg) and selenium (225 μg) for three months improved sperm cell motility and decreased lipid peroxidation in sperm cell and seminal plasma [40]. Ener et al. (2016) showed that vitamin E (600 mg α-tocopherol/day) for 12 months increased sperm count and motile sperm count in infertile male patients that underwent varicocelectomy; however, the increase in these parameters was not statistically significant [66].

Vitamin D is mainly synthesized by exposure to sunlight, which plays an important role in calcium homeostasis, bone metabolism, and cell differentiation and proliferation [67]. The importance of vitamin D has been suggested for spermatogenesis and maturation of human sperm cells, due to the expression of vitamin D receptors and metabolizing enzymes in the human testis, ejaculatory system, and mature sperm cell [41]. Serum levels of Vitamin D positively correlated with sperm cell motility and increased intracellular calcium concentration in men [43]. However, additional studies should better identify the role of vitamin D3 on male fertility. In a study by Amini et al. (2020), oral vitamin D supplementation (50,000 IU weekly for 8 weeks, and 50,000 IU per month in the following 4 weeks) showed no effects on sperm cell quality in a randomized controlled trial with infertile men [68].

Vitamin B9 or folate comprises nutritionally essential water-soluble compounds for optimal human health and development that mediates one-carbon transfer reactions, including the methylation of DNA [69]. The methylene tetrahydrofolate reductase (MTHFR) enzyme is the key enzyme for the conversion of folate to 5-methyltetrahydrofolate, the methyl group donor necessary for methionine (Met) production from homocysteine (Hcy) [69,70,71]. Folates are mainly found in dark green leafy vegetables, animal viscera, beans, whole grains, and citrus fruits [72]. Low levels of serum and seminal folate can lead to high levels of Hcy, which may induce oxidative stress, sperm DNA damage, and apoptosis, lowering sperm counts [73]. In a randomized controlled trial, Boonyarangkul et al. (2015) showed that 5 mg/day of folic acid supplement for three months improved sperm cell quality and protected against DNA damage in infertile males with semen abnormality [45]. On the other hand, three randomized controlled trials reported no significant differences regarding sperm cell volume, motility, and morphology after folate supplementation (all 5 mg/day for 12, 16, or 26 weeks) in subfertile men [74,75,76].

#### 2.1.2. Trace Elements

The trace elements selenium, calcium, copper, manganese, magnesium, sodium, potassium, and zinc are part of the seminal composition, representing key nutritional factors for proper male reproductive physiology, normal spermatogenesis, sperm maturation, motility, and function [33,77,78]. Calcium is essential for sperm cell quality, hyperactivation, the capitulation of sperm, and acrosome reaction, leading to sperm penetration into the oocyte [78]. Magnesium is involved in spermatogenesis, sperm cell motility, quality, and ejaculation, while sodium and potassium are involved in sperm motility and capacitation [78].

Zinc and selenium are the main significant elements in human semen. Selenium is an essential antioxidant micronutrient, which is critical for male reproductive tissue development and spermatogenesis, and it increases the enzymatic antioxidant activity [79]. Fish, meat products, dairy, and plants are the main dietary sources of selenium [80]. Selenium’s role is mediated by selenoproteins, the phospholipid hydroperoxide glutathione peroxidase, which is expressed by germ cells in the testis, and Selenoprotein P, a plasma protein required for selenium supply to the testis [79]. Selenium protects sperm cells against ROS [46]. Its low rates during spermatogenesis can result in abnormal sperm cells, which consequently affects semen quality, sperm cell motility, and fertility [79]. Findings of a limited number of clinical studies support selenium supplementation as a strategy to improve male reproductive physiology [47,81,82]. In a double-blind, placebo-controlled, randomized study, Safarinejad and Safarinejad (2009) administered selenium (200 µg for 26 weeks) in infertile men with idiopathic oligo-asthenoteratospermia, demonstrating improvements in semen quality [47]. Scott et al. (1998) used 100 µg/day of selenium for 12 weeks in subfertile men, resulting in increased sperm cell motility [81]. However, Hawkes et al. (2009) reported in a randomized, controlled study that selenium supplementation of 300 µg/day, for 48 weeks, did not affect testicular selenium status or semen quality in men [82]. These studies must be viewed with caution due to limitations in study design and quality [83].

Zinc is a micronutrient found in meat, wheat, and seeds. Zinc has an important function in testicular development, spermatogenesis, acrosome reaction, chemotaxis, and antioxidant action [33] since it acts as a cofactor of several enzymes involved in DNA transcription, protein synthesis, and antioxidant properties [84]. According to Colagar et al. (2009), the seminal zinc level was positively correlated with sperm cell count and normal morphology [49]. A systematic review and meta-analysis showed that infertile males have lower zinc levels in the seminal plasma compared to fertile men [48]. Moreover, in the double-blind, placebo-controlled interventional study of Wong et al. (2002), the combined ingestion of zinc (66 mg zinc sulfate) and folic acid (5 mg) for 26 weeks promoted a 74% increase in the total normal sperm cell count of subfertile men [74], suggesting that the trace element might improve sperm cell quality and male reproductive function [48].

The nutritional status of the man before conception might influence semen quality and male fertility. The deficiency of these trace elements can negatively influence the man’s reproductive health and fertility potency [78].

#### 2.1.3. Other BFCs

N-acetylcysteine is a cysteine derivative, a powerful antioxidant, and a scavenger of ROS in the treatment of OS-associated diseases [85]. Following treatment with N-acetylcysteine, ROS activity was evaluated as an approach for male infertility treatment. Ciftci et al. (2009) showed that N-acetylcysteine supplementation (600 mg/day) for three months improved the volume, motility, and viscosity of semen, probably due to reduced serum ROS production [50]. Moreover, the Safarinejad and Safarinejad (2009) study demonstrated that supplementation of N-acetylcysteine (600 mg) plus selenium (200 μg), orally and daily for 26 weeks, improved sperm cell concentration, motility, and normal morphology percent in infertile men with idiopathic oligo-asthenoteratospermia; however, no data of pregnancy occurrence were reported by authors [47].

Coenzyme Q10 is ubiquinone essential for energy production in mitochondria that also has antioxidant and membrane properties. It can be synthesized by the human body and also be obtained through salmon, tuna, beef, nuts, and seeds [86]. Three studies have evaluated the influence of coenzyme Q10 on sperm parameters in infertile men [51,52,53]. In these studies, 200 or 300 mg/day of coenzyme Q10 was supplemented for different durations (24 to 26 weeks). Safarinejad et al. (2009) results showed that 300 mg/day for 26 weeks improved sperm cell density, motility, and morphology [51]. The Balercia et al. (2009) study showed that 200 mg/day for 6 months increased the level of coenzyme Q10 and ubiquinol in seminal plasma after treatment and was effective in improving sperm cell motility [52]. Ubiquinol, a reduced form of coenzyme Q10, also improved sperm cell density, motility, and morphology [53].

Eicosapentaenoic acid (EPA; 20:5 ω-3) and docosahexaenoic acid (DHA; 22:6 ω-3) are long-chain omega-3 PUFAs obtained from the diet (e.g., fish and nuts). PUFAs are essential sperm cell membrane constituents and can influence their fluidity and integrity [14,87]. The increase in the ω-3 in the sperm plasma membrane phospholipids promotes adequate antioxidant properties, which reduce the risk of damage to sperm cells [88,89]. Lower concentrations of omega-3 have been found in the sperm cells of infertile men [55]. In a cross-sectional study by Attaman et al. (2012), the omega-3 intake was positively related to adequate sperm morphology [57]. The findings of Tang et al. (2016) suggest that omega-3 PUFA deficiency could be associated with infertility, since the infertile man had lower levels of omega-3 PUFA and greater oxidative DNA damage in sperm cell compared with the fertile men [58]. A randomized, double-blind, placebo-controlled, parallel study by Martínez-Soto et al. (2016) demonstrated that DHA (1500 mg/day) supplementation for 10 weeks increased seminal antioxidant status and decreased the percentage of sperm cells with damage [59]. Safarinejad (2010) found an association between low concentrations of omega-3 in sperm cells and poor semen quality among infertile men. These findings suggest that infertile men may benefit from omega-3 fatty acid supplementation. EPA and DHA supplementation (1.84 g/day) for 32 weeks promoted increased seminal plasma antioxidants and improved semen parameters (total sperm cell count, concentration, motility, and normal morphology) [56].

Carnitines are amines mostly provided from the diet (75%), and they can also be synthesized from essential amino acids such as lysine and methionine [90,91]. They act as co-factors in mitochondrial β-oxidation of long-chain fatty acids to enhance cellular production of energy [91]. Carnitine also protects cell membranes and DNA against ROS [92]. L-carnitine and L-acetyl-carnitine are the two major carnitine forms, which were found in the epididymal fluid and sperm cells [60]. The studies of Balercia et al. (2005) [60] and Lenzi et al. (2004) [61] evaluated the supplementation of L-carnitine and acetyl-L-carnitine, alone or in combination, for six months. In a placebo-controlled double-blind randomized trial, Lenzi et al. (2004) study showed that l-carnitine (2 g/day) and l-acetyl-carnitine (1 g/day), for six months, increased sperm cell motility in infertile males with oligo-astheno-teratozoospermia, mainly in groups with lower baseline sperm cell motility levels [61]. Balercia et al. (2005) demonstrated that, in a placebo-controlled double-blind randomized trial, the therapy with l-acetyl-carnitine (3 g/day for six months), alone or in combination with l-carnitine (l-carnitine 2 g/day plus l-acetyl-carnitine 1 g/day for six months), increased sperm cell motility; the combined therapy led to straight progressive velocity improvement after three months [60]. Protection against ROS production was also observed in the semen of men with idiopathic asthenozoospermia [60].

In a recent clinical study, Kopets et al. (2020) evaluated the effect of a dietary multi-vitamin supplement on sperm cell parameters and pregnancy rates in idiopathic male infertility with oligo-, astheno-, and teratozoospermia. Males received the supplement containing L-carnitine/L-acetylcarnitine (1990 mg), L-arginine (250 mg), glutathione (100 mg), coenzyme Q 10 (40 mg), zinc (7.5 mg), vitamin B9 (234 mcg), vitamin B12 (2 mcg), selenium (50 mcg), or placebo, daily, for six months. The percentage of spontaneous pregnancies and sperm cell quality (concentration, motility, and normal morphology) in the supplemented group was greater than in the group that received a placebo [93].

Importantly, few well-controlled clinical studies have evaluated BFCs´ potential protective effects in infertile men [83]. Two recent meta-analyses investigated the effect of antioxidant oral supplementation on male fertility. Due to the high heterogeneity of study designs, applied dose, number of participants, compounds, and evaluated parameters, further research is needed to establish more efficient methods of treating male infertility [90,94]. The review of Smits et al. (2019) concluded that antioxidant supplementation taken by subfertile men may increase the rates of pregnancy; however, the evidence is based on low-quality and small clinical trials [90]. According to Buhling et al. (2019), the meta-analysis suggests that selenium (alone or combined with N-acetylcysteine), coenzyme Q10, and the combinations of L-carnitine+acetyl-L-carnitine, folic acid+zinc, and the EPA+DHA are promising approaches for the treatment of male infertility [94].

## 3. Diet and Male Reproductive Epigenetics

### 3.1. Sperm-Specific Epigenetics

Coming from the Greek “epi” meaning over/on top, epigenetics means that molecules that are on top of the DNA structure can respond to environmental factors and can modify gene expression without changing the DNA sequence. The three main epigenetic mechanisms in mammals are: (i) DNA methylation and associated modifications, (ii) the histone/chromatin code, which consists mainly of histone variants and their post-translational modifications, and (iii) non-coding RNA [95]. These processes are cell-specific and dynamic, and they could regulate how densely specific regions of DNA are compacted, thus either inhibiting or enabling access to proteins, such as transcription factors to DNA [96].

Male gametogenesis involves intense epigenetic remodeling [23]. There are sensitive periods when environmental exposures might have amplified long-lasting effects [97]. Windows of susceptibility include pre-puberty/puberty, adulthood, and the zygote phase, which stand out as stages of development. In these phases, the epigenome is especially plastic and susceptible to disturbances induced by environmental factors such as malnutrition (Figure 1) [23].

Previously, the sperm epigenome was not of significant importance, as it was thought that, after fertilization, all epigenetic marks were erased. However, with the passing of the years and the advancement of science, studies have increasingly demonstrated that epigenetic information carried by spermatozoa can indeed be transmitted between generations [98,99,100,101,102].

Sperm cells have a unique epigenetic signature. During spermatogenesis, the sperm epigenetic profile remodeling occurs during three major steps: spermatogoniogenesis, spermatocytogenesis, and spermiogenesis [103]. A rapid expansion of the spermatogonia occurs after birth. However, after this rapid clonal expansion, the germ cells lie dormant for years until the period of puberty. From the onset of puberty through the activation of the hypothalamic-pituitary-gonadal (HPG) axis until adulthood, the process of spermatogenesis occurs in the seminiferous epithelium, and it is supported by mitotically inactive Sertoli cells [20,102].

During spermatogoniogenesis, the undifferentiated spermatogonia undergo clonal expansion through mitosis to produce spermatocytes [20]. Following this stage, specific paternal imprints are reestablished, mainly in the primordial germ cells, and end in spermatogonia [97]. Then, in spermatocytogenesis, during meiosis, most of the somatic-type histones are exchanged for testis-specific histone variants. Complete reorganization and extensive condensation of nuclear chromatin occur during spermiogenesis, leading to major replacement of most nucleosomes by protamines with histone acetylation, insertion, and removal of transition proteins [103]. The last stage of spermiogenesis is the maturation of the epididymis, and the germ cells become motile and non-coding RNA (ncRNA) mature [104]. Soon after fertilization occurs, both parental genomes are demethylated asymmetrically. However, regions of heterochromatin around centromeres largely escape this demethylation event, and with that, demethylation is not complete [20]. The epigenetic remodeling that occurs during male gametogenesis is summarized in Figure 1.

### 3.2. BFCs Epigenetic Modulation in Male Germ Cells

Although accumulating studies show that BFCs modulate several epigenetic processes in the context of cancer prevention [105], information on such effects in the context of male reproductive physiology is scarce. These dietary compounds do not directly change DNA but act on enzymes that add or remove epigenetic tags to or from DNA and histones that can activate or inhibit gene expression [103,106]. Regarding reproductive health, they could potentially impact the epigenetic landscape in male germ cells and sperm [20].

Folate is a key nutrient that impacts the epigenome [107]. It plays a critical role in 1-carbon metabolism. Folate metabolism generates the universal methyl donor S-adenosyl methionine, necessary DNA, and histone methylation. Low paternal folate intake may alter the sperm epigenome and result in adverse pregnancy outcomes [106,108]. Lambrot et al. (2013) showed that a man’s folate deficiency diet alters sperm DNA methylation at genes implicated in the development and metabolic processes [109]. However, there is very limited information on humans regarding the epigenetic modulation potential of BFCs in sperm cells, in the context of fertility.

A recent experimental study by Li et al. (2020), examined the effect of anthocyanins on spermatogenesis during puberty in male Kunming mice contaminated by cadmium (Cd). Prevalent anthocyanin C3G, found in berries, effectively protected spermatogenesis in male pubertal mice from the damage elicited by Cd via normalizing histone modification, restoring the histone to protamine exchanges in spermiogenesis, and improving the antioxidative system in the testis, subsequently alleviating apoptosis. Thus, consumption of anthocyanins can be protective against Cd-induced male pubertal reproductive dysfunction [110].

## 4. Paternal Interventions with BFCs as a Potential Epigenetic Strategy to Improve Health and Prevent Chronic Disease in the Offspring

### 4.1. Nutrition and Paternal Origins of Health and Disease (POHaD)

According to the Developmental Origins of Health and Disease (DOHaD) paradigm, adverse environmental factors operating in early life may increase the risk of chronic noncommunicable diseases in adulthood [107,111]. David Barker, a British epidemiologist, was the first to propose such a link, highlighting in his pioneer study that maternal under-nutrition during gestation can increase the risk of cardiovascular and metabolic diseases in the offspring in adulthood [112]. Further studies by Barker and Hales (1992) [113] on the “thrifty phenotype” hypothesis and studies on the “Dutch famine” cohort [112,114] expanded the knowledge of maternal experiences during critical windows of development (i.e., gestation and lactation) on later descendants health and disease risk.

Until recently, DOHaD research had focused mainly on the impact of maternal exposure because of the close connection between mother and fetus [96]. However, there is growing experimental and, to a lesser extent, clinical evidence that paternal factors during preconception also play a significant role in the metabolic health of the offspring [20]. Studies with a paternal focus have mainly shown that psychological, metabolic, and environmental factors, such as drugs, alcohol, and diet, before conception alter endocrine and metabolic functions and neurodevelopment in the offspring [115,116,117]. Evidence from these studies further suggests transmission of an epigenetic memory of past paternal exposures to the progeny through the germline [24,118,119,120,121,122]. Collectively, these studies can be positioned in a subfield of DOHaD termed POHaD [24,123].

Epigenetics has been singled out as a prominent mechanism to explain how the father´s experiences can affect offspring development [118]. As epigenetic marks are relatively stable and can be transmitted transgenerationally (F2 onwards, in the case of paternal exposures [124]), this may explain how such molecular changes would remain throughout the offspring’s adult life, resulting in the altered activities of metabolic pathways and homeostatic control processes [125,126,127].

### 4.2. Malnutrition and PoHAD

Most POHaD studies have focused on the role of paternal undernutrition on their offspring’s health [116,118,128,129,130,131]. Moderate paternal malnutrition, such as protein restriction, has been shown to impact the development of the offspring’s organs and metabolism [131,132,133]. A recent study, with young male mice receiving a low-protein (LP) diet (8.9% energy from protein), showed that F1 LP female offspring presented lower birthweight, alterations in mammary gland morphology, and expression of miR-451a, miR-200c, and miR-92a, and higher rates of breast cancer [133]. Importantly, alterations in sperm microRNA profiles of LP fathers were also identified. With a total of 16 miRNAs differentially expressed, with eight down and eight up-regulated [133].

F1 rat offspring of fathers that consumed a low protein diet (LPD; 9% casein), showed that displayed altered tissue angiotensin-converting enzyme (ACE) activity, renin-angiotensin system pathway gene expression, and vascular dysfunction [130]. In addition, similarly to F1 offspring, juvenile F2 offspring also presented alterations in growth and tissue ACE. These alterations were further accompanied by methylation of important genes such as FTO, *Mettl3*, and *Mettl14*, as well as modifications in histones, such as Hdac1, and Hdac2 [130]. In a similar study by this same research group [122], they found that sperm from LPD-fed fathers presented global hypomethylation associated with reduced testicular expression of DNA methylation enzymes Dnmt1 and Dnmt3L, as well as folate-cycle enzymes Dhfr, Mthfr, and Mtr expression. Offspring from LPD fathers became heavier, with increased adiposity, glucose intolerance, perturbed hepatic gene expression symptomatic of nonalcoholic fatty liver disease, and altered gut bacterial profiles. These data provide insight into programming mechanisms linking poor paternal diet with semen quality and offspring health [122].

According to Hajj et al. (2021) [134], paternal obesity has possible consequences on embryonic gene expression and development. *Gata6* and *Samd4b* were differentially expressed genes in embryos of high-fat diet-treated fathers [134]. *Gata6* and *Samd4b* are upregulated during adipocyte differentiation [134,135]. Thus, these genes could be involved in predisposing offspring of obese fathers to diet-induced obesity in later life [134].

Paternal obesity impact on the metabolic profile of offspring was further investigated in a male mouse model of obesity [136]. Hyperglycemia was shown in the female offspring of obese mice fathers. Importantly, methylation of the Igf2/H19 imprinting control region (ICR) was both altered in the hepatic tissue of offspring and the sperm cells of their obese fathers, suggesting that epigenetic changes in germ cells contribute to this father-offspring transmission [136].

Moreover, an experimental study [103] demonstrated that paternal obesity influences the cognitive function of offspring via epigenetic modifications in sperm cells. Paternal obesity exerted intergenerational effects on cognition in F1 offspring by increased methylation of the *Bdnf* gene promoter in the hippocampus, which could be inherited from F0 spermatozoa. *Bdnf* is a member of the neurotrophin family and plays a critical role in hippocampal neurogenesis and cognitive function [103].

A previous study by our research group [137] showed that consumption of a high-saturated fatty acid diet (60% of calories from lard), for 9 weeks during preconception, by male Sprague–Dawley rats programmed a higher risk of breast cancer in the female offspring. Interestingly, high-fat diet-treated fathers presented altered miR profiles in sperm cells [137]. It is important to highlight that clinical studies in the field of Paternal Origins of Breast Cancer are lacking [138]. Thus, an initial approach to start investigating this possibility would be to establish cohorts of daughters of obese fathers and correlate their sperm cells’ epigenetic marks with early indicators of breast cancer risk, such as mammary density during puberty and young adulthood.

Importantly, the first evidence that paternal obesity can affect a descendant’s methylation profile on imprinted genes important in embryonic growth and cancer development came from studies from the Newborn Epigenetic Study (NEST) cohort [120,121].

### 4.3. BFCs and PoHAD

Folic acid and vitamins B2, B6, and B12 are essential for one-carbon metabolism and are involved in DNA methylation. Thus, they can impact the offspring’s epigenome profile [139]. One carbon metabolism consists of the interconnected folate and methionine cycles essential for the transfer of 1C moieties required for cellular processes [140].

A recent study [141] in rats assessed whether adding a methyl donor cocktail [betaine (5 g/kg diet), choline (5.37 g/kg diet), folic acid (5.5 mg/kg diet), and vitamin B12 (0.5 mg g/kg diet)] to the paternal high fat/sucrose diet improves health status in fathers and offspring. Such paternal intervention before conception reduced energy intake and increased levels of GLP-1 and PYY hormones, which are known to reduce food intake. In addition, it improved fertility, physiological outcomes, and epigenetic and gut microbial signatures. More specifically, the methyl-donor intervention decreased the offspring hepatic expression of miR-34a and increased miR-103, miR-107, and miR-33, which are involved in lipid metabolism and the regulation of insulin sensitivity. It further elevated retroperitoneal adipose tissue expression of DNMT1, DNMT3a, and DNMT3 in the adult offspring [141]. However, when a similar dietary approach (5 g/kg diet choline chloride, 15 g/kg diet betaine, 7.5 g/kg diet methionine, 15 mg/kg diet folic acid, 1.5 mg/kg diet vitamin B12) was used on male animals on LPD, no protection was observed regarding placental physiology and epigenetic regulation [142]. The authors of this study noted that such supplementation is not a ‘quick fix’ [142].

An experimental study [143] examined whether detrimental health outcomes in offspring could be prevented by paternal micronutrient supplementation (vitamins and antioxidants). Consumption of a hypocaloric diet by male rats promoted a reduction in body weight, fertility, and overall sperm methylation, and it also increased oxidative damage to sperm cell DNA. In addition, their offspring presented reduced postnatal weight and growth but, somewhat paradoxically, increased adiposity and dyslipidemia. Interestingly, supplementing these fathers on a restricted diet with antioxidant mix (vitamin C, vitamin E, folate, lycopene, zinc, selenium, and green tea extract) normalized founder oxidative sperm DNA lesions and prevented early growth restriction, fat accumulation, and dyslipidemia in offspring. This demonstrates that paternal micronutrient supplementation during undernutrition is capable of restoring offspring metabolic health [143].

The functional analysis of the altered sperm methylome suggests that the offspring may be at increased risk for later chronic diseases, such as diabetes and cancer [144]. Another study has shown that a lifetime exposure of male mice to both folic acid deficient (0.3 mg/kg) and supplemented in excess (40 mg/kg) diets result in decreased sperm counts, adverse outcomes in their offspring, and evidence of epigenetic alterations as altered imprinted gene methylation [108]. This unbalance is a key aspect when considering supplementing future fathers with micronutrients, as both dietary deficiencies and excess may lead to the same deleterious outcomes. In a previous study by our group [145], we evaluated the potential breast cancer programming effects of selenium deficiency during preconception in male rats. Interestingly, this malnutrition condition altered mammary gland development and increased breast cancer risk in adult female offspring [145]. In addition, supplementation of male rats with selenium, during this same developmental stage, did not alter breast cancer susceptibility in offspring. One explanation would be the fact that the animals were lean and not submitted to any metabolic challenge. It has been highlighted that oxidative stress background could be a key interfering factor in selenium efficacy [146]. Thus, we [146] recently found that, when selenium was supplemented to obese fathers (a condition previously shown to program increased breast cancer risk [119]), during puberty and young adulthood, it led to amelioration of epididymal fat tissue obesity-induced oxidative stress and inflammation and reprogrammed sperm microRNA vital for spermatogenesis. This suggests that selenium supplementation during puberty and during these developmental stages could affect male physiology in the context of obesity, and it suggests that it could, potentially, positively affect offspring health [146]. These results suggest that the preconception stage is an important developmental window of opportunity to initiate nutritional preventive strategies for breast cancer, focusing on the future father´s diet during preconception.

Clinical and in vivo studies focusing on BFCs and paternal programming are still scarce. Cohorts of adult offspring of fathers submitted to specific conditions will take a long to be established, so one initial approach to expand the knowledge of BFCs in this context would be to conduct nutritional intake studies in fathers to become and correlate it with fetal development and newborn parameters. In addition, small clinical studies where interventions with BFCs shown to improve fertility parameters, as shown in Table 1, could be conducted in different populations of infertile men and men with specific metabolic conditions (i.e., obese/diabetic) to verify if alterations in sperm cell epigenetics will occur. In terms of experimental studies, one BFC approach, in this context, would be to test these same BFCs in previous models of paternal programming of specific phenotypes (diabetes, obesity, breast cancer, among others). Access to both father’s sperm and offspring adult tissue would allow more in-depth epigenetic mechanistic studies.

## 5. Conclusions

Infertility is a growing public health problem. Consumption of antioxidant micronutrients and BFCs has been highlighted as a potential strategy to protect oxidative and inflammatory damage in the male reproductive system induced by obesity, alcohol, and toxicants and thus improve spermatogenesis and fertility parameters. Despite the accumulating experimental studies showing protective effects on male reproductive health by dietary compounds, such as vitamins A, C, D, and E, as well as selenium and zinc, PUFAs, carnitines, N-acetylcysteine, and coenzyme Q10, few well-controlled and designed clinical trials are available.

Studies in the new field of PoHAD show that paternal malnutrition can alter the sperm epigenome, and this can alter fetal development and program increased risk of metabolic diseases and breast cancer in adulthood. Paternal consumption of such BFCs could not only benefit the father’s fertility/health but also their offspring’s health.

This indicates that, from a father´s perspective, preconception is a valuable window of opportunity to start nutritional interventions to maximize sperm epigenetic integrity and promote adequate fetal growth and development to prevent chronic disease in adulthood. Because the mentioned BFCs are known epigenetic modulators, they represent promising candidates for such paternal interventions (Figure 1).

## Figures and Tables

**Figure 1 nutrients-14-02150-f001:**
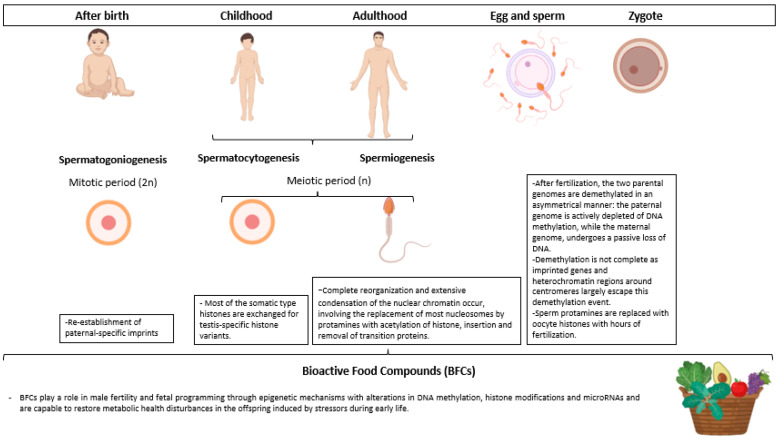
Consumption of BFCs as a potential epigenetic strategy to improve men’s reproductive health and prevent chronic disease risk in the offspring. Created with BioRender.com (accessed on 11 May 2022).

**Table 1 nutrients-14-02150-t001:** BFCs and their major outcomes on human sperm quality and function.

Nutritional Factor	Major Outcomes	References
Vitamin A	- Normal blood-testis barrier function; - Avoids germ-cell aplasia; - Fertile men have higher serum concentrations than infertile.	[36]
Vitamin C	- Improved sperm cell count, motility, and morphology; - Lower levels of vitamin C in seminal plasma of infertile men.	[37,38]
Vitamin E	- Higher live-birth rate, and a trend of better results of in vitro fertilization parameters; - Decreases the lipid peroxidation of the sperm cell and seminal plasma; - Improves sperm cell motility; - Lower levels were found in men with oligozoospermia and asthenozoospermia.	[36,39,40]
Vitamin D	- The expression of vitamin D receptors and metabolizing enzymes are marked in human testis, ejaculatory tract, and mature sperm cells; - Positive association between serum levels and sperm motility;	[41,42,43]
Vitamin B9	- Protects against DNA damage.	[44,45]
Selenium	- Protects against ROS; - Deficiency promotes sperm cell abnormalities, and affects motility and fertility;	[46,47]
Zinc	- Important for spermatogenesis: cofactor of enzymes involved in DNA transcription and protein synthesis; - Lower zinc levels in the seminal plasma of infertile men; - Increased the normal sperm cell morphology, sperm motility, and semen volume.	[48,49]
N-acetylcysteine	- Improved the volume, motility, and viscosity of sperm cells; - Increased the serum total antioxidant capacity; - Reduced the serum peroxide and oxidative stress; - Increased sperm cell concentration, motility, and percent normal morphology in infertile men.	[47,50]
Coenzyme Q10	- Improved sperm cell density, motility, and percent of normal morphology in infertile men; - Increased the seminal plasma and total antioxidant capacity.	[51,52,53,54]
Omega-3 polyunsaturated fatty acid	- Lower levels of omega-3 and greater oxidative DNA damage were found in sperm cells of infertile than infertile men; - Improved sperm cell total count, concentration, motility, and normal morphology; - Increased seminal antioxidant status and decreased the percentage of sperm cells with damage.	[55,56,57,58,59]
Carnitines	- Increased sperm cell motility; - Improved activity toward ROS in the semen.	[60,61]

Abbreviation: asthenozoospermia, impaired sperm cell motility; oligozoospermia, low sperm cell count; ROS, reactive oxygen species.

## Data Availability

Not applicable.

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
