# Peer review of "Effect of Paternal Diet on Spermatogenesis and Offspring Health: Focus on Epigenetics and Interventions with Food Bioactive Compounds"

_nutrients, 2022, doi:10.3390/nu14102150_

Round 1
Reviewer 1 Report
Effect of paternal diet on spermatogenesis and offspring health: Focus on epigenetics and interventions with micronutrients and bioactive compounds
General comments:
- This is an important area to address and you have provided a range of information about different nutrients. An area for improvement is the analysis of the literature being presented.
- At the beginning of each introduction to a vitamin, trace element or other BFC provide consistent information. For example, explain what the vitamin is, its general role and specific role in sperm. The introduction to Vitamin A is good and should be used as a template for the others
- When you speak of volume it is in reference to semen, when you speak of motility or count/concentration it is with regards to sperm to check that you are using the correct combination of words throughout
- Once you have introduced an acronym once, keep using it. Do not reintroduce again
- Be consistent with your words e.g. sperm, sperm cell, spermatozoa. Please use one word throughout for uniformity
- You mention folate in the epigenetic section, so it would be nice to include this in the earlier section too with regards to studies on folate and sperm function
- Be consistent/clear with what studies you are including. Is it meant to be a mixture of human and animal?
- Be consistent with using F1 when referring to offspring. Sometimes you say F1 and other times you say offspring
- Check gene/protein nomenclature for animals and humans
Section 1: Intro
- Line 30-32: 20-70% is a very wide range to state without any further explanation. Could you provide a brief explanation about the wide range?
- Line 33: reference for the statement of declining fertility with advanced male age.
- Line 34: provide an example of the lifestyle factors that can negatively influence sperm parameters. Also lifestyle factors encompasses adverse nutrition. Better to say “lifestyle factors such as poor nutritional intake can negatively impact”
- Line 41: “male infertile management” should be “male infertility management”. It also sounds like you are implying BFCs maintain a state of infertility. The sentence structure/meaning could be changed for more clarity e.g. “BFCs have emerged as a potential treatment for male infertility”
- Line 42/general: Define what BFCs are. I am confused if vitamins and trace elements fall under the definition of a BFC or if they are their own category.
- Line 44: what does “they” refer to?
- Line 46: “sperm reduced motility” should say “reduced sperm motility” and low number should technically be “reduced sperm count”
- Line 51: could you expand and say “non-coding RNAs, including miRNA”, rather than just miRNA
- Line 52: This sentence could be worded more clearly e.g. “epigenetic changes occur during spermatogenesis, including”
- Line 54-56: “specially during” should be changed to another word like “in particular” or “especially”. All stages of life seem to be listed so it is a redundant use of words. Also, spermatogenesis does not begin until puberty so the sentence is incorrect. This could be resolved by saying “Therefore spermatogenesis is particularly vulnerable to epigenetic alterations” and removing the stages of development that you list
- Line 63: revision should be changed to review
- Line 63-65: This sentence could be more clear and indicate if the review is focused on animals, humans or both.
Section 2: diet and male reproductive health
- Line 68: add a reference
- Line 68-70: this sentence should be reworded. SSCs do not increase productivity of sperm, they are the stem cells that sperm are derived from
- Line 70: “a healthy man” should be changed to “a fertile man”
- Line 73-77: Sub-optimal description of spermatogenesis, should be re-written to be more clear e.g. 1. Spermatogonia undergo mitosis à primary spermatocytes undergo meiosis I à 3. secondary spermatocytes undergo meiosis IIà 4. spermatids undergo spermiogenesis à 5. mature sperm capable of fertilisation. Also note that spermiogenesis is not part of spermatogenesis
Section 2.1: effects of BFCs
- Line 82: “smoke” should say smoking
- Line 86: “preventing” should say prevent
- Line 89: expand on what disturbances were found in the semen samples of infertile men from ref 19
- Line 90-91: use the acronym PUFA which you previously stated
- Line 90-92: Your earlier message in line 59-63 was that PUFAs improve outcomes following stressors but here you are saying that sperm membranes are rich in PUFAs, which is a reason why (in addition to low concentration of scavenging enzymes) sperm are particularly susceptible to oxygen induced damage. This sounds like a contradiction and needs to be explained or re-written
- Line 94: “mobility” should say motility
- Line 96: sperm physiology implies you mean only that. You could add function as well
- Line 98-99: This sentence needs to be reworded as it doesn’t make sense. Could be something like “the BFCs that improve sperm quality and functioning are summarized in table 1 with their reported outcomes”
- Line 99: You never mentioned a spermiogram until now. Either define this earlier or keep referring to it as “basic semen parameters” or “basic semen analysis”
2.1.1: vitamins
- Line 104: reference needed
- Line 106: reference needed
- Line 106-108: you have now used a rat study for the first time I believe. It needs to be made clear what studies you are using from the introduction.
- Line 108: alpha-tocopherol should not be mentioned here; this section is about vitamin C not E
- Line 111: Be consistent with using Vitamin A or ascorbic acid. Use one.
- Line 112: change “idiopathic infertile men” to men with idiopathic infertility
- Line 113: Be more specific than saying “than fertile ones”. This should say than fertile men
- Line 114-115: what do you mean by “sperm abnormality”? Do you mean abnormal sperm morphology?
- Line 116: Mobility is not used to describe sperm movement. The correct word is motility
- Line 116-117: I don’t think the study indicates that seminal ascorbic acid is associated with normal morphology à maybe that it indicates “ascorbic acid supplementation improves infertility issues in infertile men”
- Line 118-120: The way this sentence is structured makes it sound like vitamin E is a component of sperm
- Line 120-121: You are repeating what you said above in line 108-110, just in the opposite way. Remove it from above and keep it here where it is relevant to vitamin E
- Line 122: “despite not significantly increasing progressive sperm motility” needs to be placed at the end of the sentence as “did not significantly increase progressive sperm motility”
- Line 126: “during” should be “for”
- Line 135-137: I do not see the relevance of this study. If you want to include it you need to expand on how/why estrogen is important for sperm
- Line 141-142: “quality and quantity of spermograms” does not make sense. Do you mean “sperm quality” or “basic semen parameters”? Also you have now introduced another new term “spermogram”. Need to be more consistent throughout the paper
2.1.2: Trace elements
- Line 145-146: this sentence is random/ out of place
- Line 150: please change “normal ones” to fertile men. Try to be more specific at all times and not use “ones”
- Line 150-153: for refs 32 and 31 what were the doses of zinc?
- Line 151-153: this sentence should be at the end of the paragraph and reworded “Zinc supplementation improved sperm morphology, motility, volume, suggesting that the trace element might improve sperm quality and male reproductive function”
- Line 152: semen volume not just “volume”
- Line 157: reference the first sentence, why is selenium important?
- Line 159: need a reference
- Line 160: reference to selenium by its periodic element should be changed to selenium
- Line 161: reference these limited studies
2.1.3: other BFCs
- Line 165-167: “Therefore, the reduction of ROS activity in semen was evaluated as an approach for male infertility treatment” Do you mean “following treatment with N-acetylcystein, ROS activity was evaluated?”
- Line 171: Did this study see the infertile men with oligoasthenoteratospermia move into normal parameters of any sort or did they remain oligo-asthenoteratospermia, with slightly improved sperm count, motility and morphology?
- Line 177-179: which study used what duration of treatment?
- Line 183: you already defined what a PUFA is earlier
- Line 205: do you mean a multi-vitamin supplement?
3: diet and epigenetics
3.1: sperm specific
- Line 231: the compaction of DNA is not the only way in which epigenetics controls gene expression
- Line 235: effect should be plural
- Line 235-237: could you please reword this sentence and/or split into 2 sentences
- Line 248: change “in human beings” to “in humans”
- Line 249: “it IS essential” (missed the word is)
- Line 254: please explain whatH3K27me3 is
- Line 261: it is supported by Sertoli cells, not “sustained”
- Line 273: I don’t understand where “thus, the epigenome is especially plastic… conditions” comes from in this paragraph
3.2: micronutrients and BFCs
- Line 282: check typo
- Line 277-282: sounds like you contradict yourself à first saying BFCs do not directly change epigenetic marks, but then you say they could potentially impact the epigenetic landscape in sperm
- Line 291-298: Most of this belongs in a section earlier. Only the histone information is relevant
4.
4.1: Nutrition and PoHAD
- Line 313: preconception typo
- Line 314: are the studies in animals or humans or both?
- Line 315: Psychic is not the correct word to be using. Do you mean psychological?
- Line 322: Define transgenerationally
- Line 325: is this using human studies, animal studies or both?
4.2: Malnutrition and PoHAD
- Line 336: F1 LP female offspring sounds more appropriate than daughters
- Line 337: Please explain what miR is
- Line 341: define ACE and RAS
- Line 344-345: Check gene nomenclature for mice (italicise all, capitalise first letter for mice genes)
- Line 358-360: check gene nomenclature
- Line 371: define intergenerational
4.3: micronutrients and bioactive food
- Line 401: weather is a typo à whether
- Line 403: this sentence needs to be reworded for clarity
- Line 407: what quantities were the antioxidant mixtures?
- Line 417: what is meant by u-shaped pattern?
- Line 421-425: this could be reworded to be more clear e.g. supplementation with selenium in male mice reduces the risk of breast cancer in offspring
Conclusion:
- Line 436: can alter THE sperm (typo)
- Line 438: this indicates* that (needs plural)
Table 1:
- is this human only?
- The table needs to be more clearly defined. I can’t tell where one factor ends and the next starts
- I am confused if the arrows are referring to the nutritional factor or the major outcome variables
- It would be nice to compare the amounts of the nutritional factor in the studies if it was an intervention. If an intervention study, state the amount of the vitamin
- For vitamin C- you say “morphology” but you do not say what is the outcome (improved?) can you please add the outcome
- Define what oligo- and astheno- zoospermia are beneath the table, and ROS
- For zinc:
- you say increase viscosity but there is no indication if this is a positive or negative thing
- you also say increase motility of semenà this should say sperm
Figure 1.
- you mention phytochemicals for the first time here. You should therefore mention it in the paper as well
- the figure has grammatical errors, can the authors please revise grammar/spelling
- gene symbols should be italicised and capitalised if talking about humans, which the diagram appears to be
- formatting issues with spaces between words
Author Response
We thank Reviewer 1 for the careful analysis of our manuscript and relevant recommendations that improved very much its quality. Please find our responses to all your questions below. We further inform that modifications in the manuscript are highlighted in yellow background.
REVIEWER 1
Open Review
General comments:
- This is an important area to address and you have provided a range of information about different nutrients. An area for improvement is the analysis of the literature being presented.
- At the beginning of each introduction to a vitamin, trace element or other BFC provide consistent information. For example, explain what the vitamin is, its general role and specific role in sperm. The introduction to Vitamin A is good and should be used as a template for the others
- When you speak of volume it is in reference to semen, when you speak of motility or count/concentration it is with regards to sperm to check that you are using the correct combination of words throughout
- Once you have introduced an acronym once, keep using it. Do not reintroduce again
- Be consistent with your words e.g. sperm, sperm cell, spermatozoa. Please use one word throughout for uniformity
- You mention folate in the epigenetic section, so it would be nice to include this in the earlier section too with regards to studies on folate and sperm function
- Be consistent/clear with what studies you are including. Is it meant to be a mixture of human and animal?
- Be consistent with using F1 when referring to offspring. Sometimes you say F1 and other times you say offspring
- Check gene/protein nomenclature for animals and humans
Section 1: Intro
- Line 30-32: 20-70% is a very wide range to state without any further explanation. Could you provide a brief explanation about the wide range?
R: These mean values were taken from Table 1 of the article by Agarwal et al (2015) where men were interviewed from all continents of the world who reported any of the infertility diagnoses or were undergoing in vitro fertility treatment in specialized clinics. We have now introduced two additional references that support these figures. Please see page 3.
- Line 33:reference for the statement of declining fertility with advanced male age.
R: This sentence was taken from the article "Fertility and the aging male" by Harris et al (2011), allocated in the text as reference number 5. Please see page 3.
- Line 34:provide an example of the lifestyle factors that can negatively influence sperm parameters. Also lifestyle factors encompass adverse nutrition. Better to say “lifestyle factors such as poor nutritional intake can negatively impact”
R: Thank you for this remark. The modification has been made to the text. Please see page 3.
- Line 41: “male infertile management” should be “male infertility management”. It also sounds like you are implying BFCs maintain a state of infertility. The sentence structure/meaning could be changed for more clarity e.g. “BFCs have emerged as a potential treatment for male infertility”
R: Thank you for this remark. The modification has been made to the text. Please see page 3.
- Line 42/general: Define what BFCs are. I am confused if vitamins and trace elements fall under the definition of a BFC or if they are their own category.
R: The definition of BFCs was added in the introduction as reviewer suggested. Bioactive compounds can be defined as nutrients and non-nutrients present in the food matrix (vegetal and animal sources) that can produce physiological effects beyond their classical nutritional properties. Thus, vitamins and trace elements are bioactive food components and also nutrients, whereas polyphenolic compounds are also bioactive compounds but non-nutrient compounds. The title and abstract and several subitems of the manuscript were adjusted to make this point more clear. Please see pages 1, 2, 3, 5 and table 1.
- Line 44:what does “they” refer to?
R: We are referring to bioactive food compounds. It is now explicit in the text. Please see page 3.
- Line 46:“sperm reduced motility” should say “reduced sperm motility” and low number should technically be “reduced sperm count”
R: Thank you for your comment. The change was made.
- Line 51:could you expand and say “non-coding RNAs, including miRNA”, rather than just miRNA
R: Thank you for your comment. The change was made.
- Line 52:This sentence could be worded more clearly e.g. “epigenetic changes occur during spermatogenesis, including”
R: The correction was made in the text.
- Line 54-56: “specially during” should be changed to another word like “in particular” or “especially”. All stages of life seem to be listed so it is a redundant use of words. Also, spermatogenesis does not begin until puberty so the sentence is incorrect. This could be resolved by saying “Therefore spermatogenesis is particularly vulnerable to epigenetic alterations” and removing the stages of development that you list
R: We agree with the reviewer. We have adjusted the sentence accordingly. Please see page 15.
- Line 63: revision should be changed to review
R: The correction was made in the text.
- Line 63-65: This sentence could be more clear and indicate if the review is focused on animals, humans or both.
R: Thank you for this remark. The new sentence became “Thus, this review will focus on epigenetics and interventions with food bioactive compounds in the paternal diet on spermatogenesis and offspring health based on clinical and in vivo studies”. Please see page 4.
Section 2: diet and male reproductive health
- Line 68:add a reference
R: The reference was added.
- Line 68-70:this sentence should be reworded. SSCs do not increase productivity of sperm, they are the stem cells that sperm are derived from.
R: The sentence was reworded: “After puberty, spermatogonial stem cells (SSCs) provide the foundation of sperm cell, a process that persists throughout the majority of a male’s lifetime.”
- Line 70:“a healthy man” should be changed to “a fertile man”
R: As suggested, “a healthy man” was changed to “fertile man”.
- Line 73-77:Sub-optimal description of spermatogenesis, should be re-written to be more clear e.g. 1. Spermatogonia undergo mitosis à primary spermatocytes undergo meiosis I à 3. secondary spermatocytes undergo meiosis IIà 4. spermatids undergo spermiogenesis à 5. mature sperm capable of fertilisation. Also note that spermiogenesis is not part of spermatogenesis
R: The description of spermatogenesis was re-written as reviewer suggested.
Section 2.1: effects of BFCs
- Line 82:“smoke” should say smoking
R: The correction was made in the text.
- Line 86:“preventing” should say prevent
R: The correction was made in the text.
- Line 89:expand on what disturbances were found in the semen samples of infertile men from ref 19
R: The disturbances found in the reference were expanded.
- Line 90-91:use the acronym PUFA which you previously stated
R: The correction was made in the text.
- Line 90-92:Your earlier message in line 59-63 was that PUFAs improve outcomes following stressors but here you are saying that sperm membranes are rich in PUFAs, which is a reason why (in addition to low concentration of scavenging enzymes) sperm are particularly susceptible to oxygen induced damage. This sounds like a contradiction and needs to be explained or re-written
R: The sentence has been rewritten to make the message clear. The sperm plasma membranes are constituted of PUFAS, which makes them susceptible to oxygen-induced damage. However, the increase in omega-3 PUFAs at the expense of n-6 PUFAs promote adequate antioxidant properties and reduce the risk of damage to sperm cells.
“Sperm cell plasma membranes are constituted of phospholipids and their cytoplasm contains low concentrations of scavenging enzymes (...)”.
And "Eicosapentaenoic acid (EPA; 20:5 ω-3) and docosahexaenoic acid (DHA; 22:6 ω-3) are long-chain omega-3 PUFAs obtained from the diet (e.g. fish and nuts). Omega-3 PUFAs are essential sperm cell membrane constituents and can influence their fluidity and integrity [7,64]. The increase of the ω-3 in the sperm plasma membrane phospholipids promotes adequate antioxidant properties which reduce the risk of damage to sperm cells [64]. Lower concentrations of omega-3 have been found in sperm cells of infertile men [39]."
- Line 94:“mobility” should say motility
R: The correction was made in the text.
- Line 96:sperm physiology implies you mean only that. You could add function as well
R: Thank you for the correction. The text has been corrected accordingly.
- Line 98-99:This sentence needs to be reworded as it doesn’t make sense. Could be something like “the BFCs that improve sperm quality and functioning are summarized in table 1 with their reported outcomes”
R: The correction was made in the text.
- Line 99:You never mentioned a spermiogram until now. Either define this earlier or keep referring to it as “basic semen parameters” or “basic semen analysis”
R: Thank you for this remark. “Basic semen parameters” was used in the text.
2.1.1: vitamins
- Line 104: reference needed
R: The reference has been added.
- Line 106: reference needed
R: The reference has been added.
- Line 106-108: you have now used a rat study for the first time I believe. It needs to be made clear what studies you are using from the introduction.
R: We are thankful for the comment. The manuscript revision has been made to make it clear. We inform in the Introduction section that we cover human and in vivo studies in the present review article. Please see
- Line 108: alpha-tocopherol should not be mentioned here; this section is about vitamin C not E
R: The correction was made accordly.
- Line 111:Be consistent with using Vitamin C or ascorbic acid. Use one.
R: “Vitamin C” was chosen and cited throughout the text.
- Line 112:change “idiopathic infertile men” to men with idiopathic infertility
R: The correction was made in the text.
- Line 113:Be more specific than saying “than fertile ones”. This should say than fertile men
R: The correction was made in the test.
- Line 114-115:what do you mean by “sperm abnormality”? Do you mean abnormal sperm morphology?
R: We mean “abnormal sperm cell morphology” and the correction was made in the text.
- Line 116:Mobility is not used to describe sperm movement. The correct word is motility
R: We are thankful for the reviewer correction. The word “motility” was used to describe sperm cell movement.
- Line 116-117:I don’t think the study indicates that seminal ascorbic acid is associated with normal morphology à maybe that it indicates “ascorbic acid supplementation improves infertility issues in infertile men”
R: Thank you. The correction was carried out as indicated.
- Line 118-120:The way this sentence is structured makes it sound like vitamin E is a component of sperm
R: Thank you. The sentence has been rewritten.
- Line 120-121: You are repeating what you said above in line 108-110, just in the opposite way. Remove it from above and keep it here where it is relevant to vitamin E
R: The sentence was removed from line 108-110 and kept at vitamin E paragraph, where it is relevant.
- Line 122:“despite not significantly increasing progressive sperm motility” needs to be placed at the end of the sentence as “did not significantly increase progressive sperm motility”
R: The correction was made in the text according to the reviewer's suggestion.
- Line 126:“during” should be “for”
R: The correction was made in line 126.
- Line 135-137:I do not see the relevance of this study. If you want to include it you need to expand on how/why estrogen is important for sperm
R: This study has been removed.
- Line 141-142:“quality and quantity of spermograms” does not make sense. Do you mean “sperm quality” or “basic semen parameters”? Also you have now introduced another new term “spermogram”. Need to be more consistent throughout the paper.
R: Thank you for this remark. We mean “sperm cell quality” in line 141-142. The correction was made in the text.
2.1.2: Trace elements
- Line 145-146:this sentence is random/ out of place
R: Thank you for this remark. This sentence (line 145-146) was removed.
- Line 150: please change “normal ones” to fertile men. Try to be more specific at all times and not use “ones”
R: Thank you for this remark. “Normal ones” was changed to “fertile men”, as reviewer suggested.
- Line 150-153:for refs 32 and 31 what were the doses of zinc?
R: References 31 and 32 are not nutritional interventions, therefore, zinc supplementation was not performed. Ref 31 (Zhao et. al. 2016) is a systematic review and meta-analysis, and Ref 32 (Colagar, et al. 2009) is an observational study.
- Line 151-153:this sentence should be at the end of the paragraph and reworded “Zinc supplementation improved sperm morphology, motility, volume, suggesting that the trace element might improve sperm quality and male reproductive function”.
R: The sentence (line 151-153) was placed at the end of the paragraph and reworded, as reviewer suggested.
- Line 152:semen volume not just “volume”
R: The correction was made in the text.
- Line 157:reference the first sentence, why is selenium important?
R: Thank you for the remark. An explanation of the selenium importance was given.
- Line 159:need a reference
R: The reference was added.
- Line 160:reference to selenium by its periodic element should be changed to selenium
R: The correction was made in the text.
- Line 161: reference these limited studies
R: The studies were referenced accordingly.
2.1.3: other BFCs
- Line 165-167:“Therefore, the reduction of ROS activity in semen was evaluated as an approach for male infertility treatment” Do you mean “following treatment with N-acetylcystein, ROS activity was evaluated?”
R: We are thankful to valuable comments of the reviewer. The correction was made in the text.
- Line 171:Did this study see the infertile men with oligoasthenoteratospermia move into normal parameters of any sort or did they remain oligo-asthenoteratospermia, with slightly improved sperm count, motility and morphology?
R: The supplementation improved semen parameters in infertile men, however, this increase was not enough to move into normal parameters. No data of pregnancy occurence were reported by authors.
- Line 177-179:which study used what duration of treatment?
R: The duration of treatment and dose of coenzyme Q10 were identified in which study.
- Line 183:you already defined what a PUFA is earlier
R: The correction was made in the text.
- Line 205:do you mean a multi-vitamin supplement?
R: The term “multi-vitamin supplement” was used in the text.
3: diet and epigenetics
3.1: sperm specific
- Line 231:the compaction of DNA is not the only way in which epigenetics controls gene expression
R: Yes, we appreciate your comments, but for this reason we wrote "could regulate" instead of stating as the only mechanism "regulates..."
- Line 235:effect should be plural
R: The modification was made.
- Line 235-237:could you please reword this sentence and/or split into 2 sentences
R: Thank you for this remark. The new sentence became “Windows of susceptibility include pre-puberty/puberty, adulthood, and the zygote phase, that stand out as stages of development. In these phases, the epigenome is especially plastic and susceptible to disturbances induced by environmental factors such as malnutrition”.
As indicated by Referee 2, we now R: We have removed information on intrauterine stage and focused on spermatogenesis, to be in line with the objective of the review article.
- Line 248:change “in human beings” to “in humans”
R: The modification was made.
- Line 249:“it IS essential” (missed the word is)
R: The modification was made.
- Line 254:please explain whatH3K27me3 is
R: It is a mark that indicates the tri-methylation of lysine 27 on histone H3 protein.The new sentence was “...via loss of histone H3K27me3…”
- Line 261:it is supported by Sertoli cells, not “sustained”
R: The modification was made.
- Line 273:I don’t understand where “thus, the epigenome is especially plastic… conditions” comes from in this paragraph
R: The sentence was removed from the text.
3.2: micronutrients and BFCs
- Line 282: check typo
R: Thank you. The correction was made.
- Line 277-282:sounds like you contradict yourself à first saying BFCs do not directly change epigenetic marks, but then you say they could potentially impact the epigenetic landscape in sperm
R: Thank you. The new sentence is “These dietary compounds do not directly change DNA, but act on enzymes that add or remove epigenetic tags to or from DNA and histones that can activate or inhibit gene expression”.
- Line 291-298:Most of this belongs in a section earlier. Only the histone information is relevant
R: Thank you. The new sentence is “A recent study by Li et al (2020), examined the effect of anthocyanins on spermatogenesis during puberty in male Kunming mice contaminated by cadmium (Cd). Prevalent anthocyanin C3G found in berries effectively protected spermatogenesis in male pubertal mice from the damage elicited by Cd via normalizing histone modification, restoring the histone to protamine exchanges in spermiogenesis, and improving the antioxidative system in the testis, subsequently alleviating apoptosis. Thus, consumption of anthocyanins can be protective against Cd-induced male pubertal reproductive dysfunction”.
4.
4.1: Nutrition and PoHAD
- Line 313:preconception typo
R: Thank you. The correction was made.
- Line 314:are the studies in animals or humans or both?
R: These studies are human clinical studies and animal studies (in vivo).
- Line 315:Psychic is not the correct word to be using. Do you mean psychological?
R: Thank you. The correction was made.
- Line 322:Define transgenerationally
R: The definition is F2 onwards in case of paternal exposures. This is now referenced in the text,
- Line 325:is this using human studies, animal studies or both?
R: These studies are human clinical studies and animal studies (in vivo).
4.2: Malnutrition and PoHAD
- Line 336:F1 LP female offspring sounds more appropriate than daughters
R: Thank you. The correction was made.
- Line 337:Please explain what miR is
R: Was replaced by the word microRNA.
- Line 341:define ACE and RAS
R: Angiotensin-converting enzyme - ACE and renin–angiotensin system (RAS). Were placed in the text.
- Line 344-345: Check gene nomenclature for mice (italicise all, capitalise first letter for mice genes)
R: Thank you. The correction was made.
- Line 358-360:check gene nomenclature
R: Thank you. The correction was made.
- Line 371:define intergenerational
R: The correct word in the text is “transgenerational”. The correction was made.
4.3: micronutrients and bioactive food
- Line 401:weather is a typo à whether
R: Thank you. The correction was made.
- Line 403:this sentence needs to be reworded for clarity
R: Thank you for the suggestion. The new sentence is “Consumption of a hypocaloric diet by male rats promoted a reduction in body weight, fertility, increased oxidative damage to sperm cell DNA and reduced overall sperm methylation”.
- Line 407:what quantities were the antioxidant mixtures?
R: According to supplementary table 1 of McPherson et al (2016), the quantities of antioxidant mixtures are: Selenium premix (4.4% Se) 0.14g, LycoRed (20% Lycopene as supplied) 0.19g, Vitamin E (50% m/m) 1.35g, Folic acid (90%) 0.06g, Zinc sulphate monohydrate (35.8% Zn) 2.79g, Green tea extract 0.95g.
- Line 417:what is meant by u-shaped pattern?
R: This U-shaped pattern was the author's way of explaining the 2 extreme points (folic acid deficiency and supplementation; low point followed by high point), resulting in a decrease (another low point on the curve) of ,”sperm counts, adverse outcomes in their offspring, and evidence of epigenetic alterations as altered imprinted gene methylation” resembling the drawing of an upside down U.
- Line 421-425:this could be reworded to be more clear e.g. supplementation with selenium in male mice reduces the risk of breast cancer in offspring
R: We appreciate the suggestion. We have rewritten this section to make it more clear and also added a recent published article by our group that improve discussion of this particular contexto.
Reference: Pascoal, G.d.F.L.; Novaes, G.M.; Sobrinho, M.d.P.; Hirayama, A.B.; Castro, I.A.; Ong, T.P. Selenium Supplementation during Puberty and Young Adulthood Mitigates Obesity-Induced Metabolic, Cellular and Epigenetic Alterations in Male Rat Physiology. Antioxidants 2022, 11, 895. https://doi.org/10.3390/ antiox11050895
Conclusion:
- Line 436:can alter THE sperm (typo)
R: Thank you. The correction was made.
- Line 438: this indicates* that (needs plural)
R:Thank you. The correction was made.
Table 1:
- is this human only?
R: The table contains the summary of BFCs outcomes described in 2.0 topic. The table title has been changed to contain this information. “Bioactive food compounds (BFCs) and their major outcomes on human sperm quality and functioning.
”
- The table needs to be more clearly defined. I can’t tell where one factor ends and the next starts.
R: We are thankful for the reviewer correction. The table has been edited to be clearly defined.
- I am confused if the arrows are referring to the nutritional factor or the major outcome variables
R: The arrows have been removed and an outcome description has been added.
- It would be nice to compare the amounts of the nutritional factor in the studies if it was an intervention. If an intervention study, state the amount of the vitamin.
R: Table 1 contains a summary of the main outcomes of each nutrient. Doses and intervention period are better described in the text.
- For vitamin C- you say “morphology” but you do not say what is the outcome (improved?) can you please add the outcome
R: To make the table information clearer, the arrows have been removed and an outcome description has been added.
- Define what oligo- and astheno- zoospermia are beneath the table, and ROS
R: We are thankful for the reviewer correction. The list of abbreviations has been added below the table.
- For zinc:
- you say increase viscosity but there is no indication if this is a positive or negative thing
R: The increase of viscosity is a positive thing. To make the information clearer, the arrows have been removed and an outcome description has been added.
- you also say increase motility of semenà this should say sperm
R: The correction has been made.
Figure 1.
- you mention phytochemicals for the first time here. You should therefore mention it in the paper as well
R: The new sentence is “BFC’s play a role in male fertility and fetal programming through epigenetic mechanisms with alterations in DNA methylation, histone modifications and microRNAS and is capable to do restoration of metabolic health of offspring induced by stressors during early life”.
- the figure has grammatical errors, can the authors please revise grammar/spelling
R: Thanks for the observation. All errors have been corrected.
- gene symbols should be italicised and capitalised if talking about humans, which the diagram appears to be
R: Thank you. The correction was made.
- formatting issues with spaces between words
R: Thank you. The correction was made.
Reviewer 2 Report
Thank you for the opportunity to review this work. The manuscript under this review concerns epigenetics and interventions with micronutrients and bioactive compounds in the paternal diet on spermatogenesis and offspring health.
The study is intriguing, but due to its character as a review, it needs an improvement in its bibliographic search and to show more relevant articles in the topic shown.
Abstract
The presentation of the abstract is clear.
- Introduction
The introduction needs to expand the literature search for more relevant articles in support of the considerations the authors present.
Line 28: The first time the World Health Organization is named, use the acronym; for example (WHO).
Lines 33-36: I suggest the authors to support this sentence with so many factors affecting sperm parameters with more and more updated references, such as PMID: 30196939, PMID: 30935865, PMID: 34884971 and PMID: 30149588.
Lines 36-40: What sense does this sentence make with the previous one? How does nutrition imbalance affect sperm quality? I suggest you change the order of these two sentences in the paragraph or rewrite it to give it a better meaning.
Lines 41-43: I suggest that you support this sentence with more relevant references, if any.
Line 45: Oxidative stress (OS) damage or rather damage due to reactive oxygen species (ROS)?
Line 50: The first time you name DNA in the manuscript, indicate what it means; deoxyribonucleic acid.
Line 55: "especially during the intrauterine pre-puberty/puberty..." Intrauterine? Are the authors sure to explain this concept by talking about spermatogenesis in the manuscript?
- 2. Diet and male reproductive health
Line 78: which hormonal and signalling pathways? I suggest the authors name them and give details for a better understanding, justifying with relevant references.
2.1. Effects of BFCs on male gametogenesis
Line 79: The title of this subsection could be "spermatogenesis" instead of "male gametogenesis" to unify the terminology used throughout the manuscript.
Lines 81-84: As suggestions shown above, I suggest the authors add more bibliographic citations to this sentence due to the large number of adverse environmental factors listed by the authors.
2.1.1. Vitamins
Lines 104-105: In the introductory sentence on Vitamin A, I suggest that it be supported by a reference.
Line 111: Would it not be better to place ascorbic acid on line 104, which is where this vitamin occurs?
Lines 121-124: To unify results, I suggest indicating the name of the author of this double-blind, placebo-controlled randomised study, which is Keskes-Ammar et al (2003). Thus, in subsequent sections of the manuscript.
Lines 124-127: Similar suggestion as above.
Line 130: The order of the references is not correct. I suggest that they be modified, since in table 1 there are references that are not ordered in the order in which they appear and affect the manuscript. Please correct this.
Lines 131-135: Similar suggestion as above. When discussing the importance of Vitamin D in spermatogenesis and maturation, more relevant references should be provided.
Lines 139-142: This study should be presented in a different form, indicating author, year and in comparison, with the previous study, I suggest the authors to rewrite this presentation of the study.
2.1.2. Trace elements
In this sub-section of trace elements, only Zinc and Selenium are presented, but what do the authors think about the relevance of Iron and Calcium, for example, in seminal composition?
Line 144-145: Which reference do the authors think may be relevant to support this statement?
Lines 148-156: On these lines, please indicate the studies showing these results; name, year and type of study. In addition, I suggest that the authors rewrite this section for a better understanding of what they want to explain about Zinc.
Lines 160-162: Which studies? No studies are presented in the text. Please expand on this information.
2.1.3. Other BFCs
Lines 164-171: In this paragraph, please present the studies as suggested in the previous sections.
Lines 172-174: Add references in the presentation to Coenzyme Q10 and those that indicate the synthesis of Coenzyme Q10 and the foods where it is abundant, as explained in the manuscript.
Lines 174-180: The presentation of these studies with Coenzyme Q10 should be unified. I suggest the authors rewrite this paragraph.
Lines 182-185: I need references to be added in these two sentences to indicate the presence of PUFAs, as well as their relevant role in the sperm membrane.
Lines 186-191: The authors present only one study in this review in relation to sperm quality and low omega-3 concentrations. Are they confident that they have conducted a commensurate search to present these results? I suggest reviewing PMID: 21219381, PMID: 25405715, PMID: 22416013 and PMID: 27792396.
Lines 192-194: Please support these two sentences with relevant references.
Lines 197-203: I suggest that you present the Balercia and Lenzi studies in a different way, organising the information they indicate.
Lines 212-221: The authors discuss the meta-analyses of Buhling and Smits, but only present the results found in Buhling's meta-analysis. What results are found in Smits' meta-analysis? Similar to Buhling's? Different? Explain them and add to the meaning of this paragraph.
- Diet and male reproductive epigenetics
3.1. Sperm-specific epigenetics
Line 229: The first time RNA appears in the text, explain its meaning as ribonucleic acid (RNA).
Line 235: "Intrauterine" in "male gametogenesis" is a term that is not in line with the objective of the review presented by the authors. I suggest changing or modifying the meaning of the paragraph where this term appears.
Lines 242-244: The authors present just two studies showing the advancement of science and epigenetic information associated with spermatogenesis. I suggest to expand this information with other relevant studies; PMID: 33835194, PMID: 24908726, PMID: 29717022
Lines 248-251: "During the intrauterine period?" I suggest the authors modify the meaning of this sentence, as the manuscript should focus on spermatogenesis.
Lines 251-261: In this sequence of sentences, I do not perceive that an extensive literature search has been carried out to justify the use of these references presented.
Line 259: Please use the meaning of HPG the first time it is used in the manuscript.
Lines 262-269: I suggest that the authors reposition the quotations used for these phrases [8,55,57] in such a way that it is understood for which phrase is the most appropriate, as the way they are presented seems confusing for the reader.
Line 270: What does ncRNA mean? Show the meaning of non-coding RNA the first time it is used in the manuscript as non-coding RNA (ncRNA). Where is the supporting reference for this sentence?
Line 275: I suggest the authors to modify or change the position of figure 1 at the end of this paragraph because it has a better interpretation in this paragraph.
3.2. Micronutrients and BFCs epigenetic modulation in male germ cells
Line 283: The first sentence is relevant and important enough to support it with a reference. Consider PMID: 16251634
Why, if folate is so important in epigenetic modulation related to male gametogenesis, is it not explained in the previous subsection 2.1. Effects of BFCs on male gametogenesis? Justify this aspect and modify.
Line 291: What study? what author? It is relevant to show this study carried out in mice that serves to relate it to spermiogenesis in humans based on epigenetics and to show a unified presentation of the works with the authors throughout the manuscript.
- Paternal interventions with BFCs as a potential epigenetic strategy to improve health and prevent chronic disease in the offspring
4.1. Nutrition and Paternal Origins of Health and Disease (PoHAD)
Lines 302-304: The authors present only one reference that supports this statement about the DOHaD paradigm?
Lines 318-318: What several studies do the authors suggest in this sentence? The authors only present the Yeshurun ​​study from 2017, which is a mouse study. I suggest authors modify this sentence or search for references that are relevant to this topic. Same suggestion as presented in line 319.
Lines 320-324: I suggest rewriting this paragraph by adding relevant references, as suggested above.
Lines 325-327: I suggest you delete this sentence, because it is not relevant to present the following sections at this point.
Delete the spaces between section 4.1 and 4.2 (lines 328-330).
4.2. Malnutrition and PoHAD
If this subsection only presents studies in mice, it should be rewritten to show the relevance and importance of conducting studies in humans.
Lines 332-333: Which majority? The authors keep referencing Yeshurun 2017 study which is not a systematic review and was carried out on mice. Please rewrite this presentation.
Lines 334-339: Shouldn't the authors focus more on alterations in sperm miR profiles rather than alterations in mammary gland morphology? Are there no human studies? I suggest the authors carry out a bibliographic search on this topic to retrieve relevant studies in humans and rewrite these lines.
Line 341: What does ACE and RAS pathway mean? Name the meaning of these acronyms the first time they appear in the manuscript.
Lines 353-362: Studies carried out in humans. Perfect. You should consider separating between human and mouse studies if you decide to keep this section for better understanding.
Lines 363-367: The authors join the above studies with the study by Wu et al. which is done on mice. I suggest rewriting this section 4.2 so as not to conflate the results found in mice and human studies.
Lines 369-374: A 2018 study is not a recent study. In addition, it is another work presented in mice.
Lines 375-379: The research group's study is also carried out on mice. How could it be extrapolated to future human studies? Justify these suggestions in this section.
4.3. Micronutrients and Bioactive Food Compounds and PoHAD
I recommend the authors in this subsection to present the limitations of the scarcity of human studies and future lines of research to overcome them.
Lines 386-400: The authors present only one study in rats. Do you think that with only one study it is relevant to present data? I suggest that they add more information on macronutrients and BFCs as suggested in previous sections and their relationship to epigenetics and spermatogenesis. The data presented by the authors are insufficient.
Lines 401-411: This study carried out by McPherson et al. It has also been done in mice. Consider presenting human work or showing limitations to the scarcity of human work in relation to the micronutrients and BFCs presented in this section for the future.
Lines 413-425: Similar to the above considerations.
- Conclusions
The authors should separate the key points of the conclusions in different paragraphs for the different results presented.
The limitations that exist in the review presented by the authors in human studies that show the relationship between epigenetics and spermatogenesis and infertility problems and how they could be improved with food should be present. I believe that these aspects remain to be demonstrated.
Author Response
We thank Reviewer 2 for the careful analysis of our manuscript and relevant recommendations that improved very much its quality. Please find our responses to all your questions below. We further inform that modifications in the manuscript are highlighted in yellow background.
R2
Introduction
The introduction needs to expand the literature search for more relevant articles in support of the considerations the authors present.
Line 28: The first time the World Health Organization is named, use the acronym; for example (WHO).
R: Thank you. The insertion of the acronym was carried out.
Lines 33-36: I suggest the authors to support this sentence with so many factors affecting sperm parameters with more and more updated references, such as PMID: 30196939, PMID: 30935865, PMID: 34884971 and PMID: 30149588.
R: Thanks for the suggestion. The factors have been expanded in the text and references have been added in the text.
Lines 36-40: What sense does this sentence make with the previous one? How does nutrition imbalance affect sperm quality? I suggest you change the order of these two sentences in the paragraph or rewrite it to give it a better meaning.
R: Excellent suggestion. The paragraph has been rewritten.
Lines 41-43: I suggest that you support this sentence with more relevant references, if any.
R: Thanks for the suggestion. Some references have been added in the text.
Line 45: Oxidative stress (OS) damage or rather damage due to reactive oxygen species (ROS)?
R: The modification was made.
Line 50: The first time you name DNA in the manuscript, indicate what it means; deoxyribonucleic acid.
R: The modification was made.
Line 55: "especially during the intrauterine pre-puberty/puberty..." Intrauterine? Are the authors sure to explain this concept by talking about spermatogenesis in the manuscript?
We agree with the important point raised by R2 as well as R1. We agree that sperm cell formation only starts at puberty. However, from a developmental epigenetic perspective, earlier stages (i.e. in utero) have an important impact on the initial epigenetic pattern of primordial germ cells that can influence subsequent sperm cell epigenetic signatures. This is now informed in the text as: “Importantly, stages before the beginning of spermatogenesis are key for later influencing epigenetic signatures in sperm cells (refs)”. After this information, we have maintained the original text commenting on in utero epigenetic modulation of primordial germ cells.
- Diet and male reproductive health
Line 78: which hormonal and signalling pathways? I suggest the authors name them and give details for a better understanding, justifying with relevant references.
R: We have expanded this information accordingly.
2.1. Effects of BFCs on male gametogenesis
Line 79: The title of this subsection could be "spermatogenesis" instead of "male gametogenesis" to unify the terminology used throughout the manuscript.
R: The title of 2.1 section has been changed according to the revisor's suggestion.
Lines 81-84: As suggestions shown above, I suggest the authors add more bibliographic citations to this sentence due to the large number of adverse environmental factors listed by the authors.
R: More bibliographic citations have been added to this sentence.
2.1.1. Vitamins
Lines 104-105: In the introductory sentence on Vitamin A, I suggest that it be supported by a reference.
R: The reference was added.
Line 111: Would it not be better to place ascorbic acid on line 104, which is where this vitamin occurs?
R: The foods in which vitamin C is found have been added, as can be seen in the text " Vitamin C, also known as ascorbic acid, is mainly found in fruits and vegetables and has the functionality to reduce DNA damage directly by scavenging free radicals and decreasing lipid peroxidation [48]. "
Lines 121-124: To unify results, I suggest indicating the name of the author of this double-blind, placebo-controlled randomised study, which is Keskes-Ammar et al (2003). Thus, in subsequent sections of the manuscript.
R: The author name of the study was added to unify results.
Lines 124-127: Similar suggestion as above.
R: The author name of the study was added to unify results.
Line 130: The order of the references is not correct. I suggest that they be modified, since in table 1 there are references that are not ordered in the order in which they appear and affect the manuscript. Please correct this.
R: The references’ correction was made accordly.
Lines 131-135: Similar suggestion as above. When discussing the importance of Vitamin D in spermatogenesis and maturation, more relevant references should be provided.
R: The reference for vitamin D discussion and importance was added.
Lines 139-142: This study should be presented in a different form, indicating author, year and in comparison, with the previous study, I suggest the authors to rewrite this presentation of the study.
R: The study description has been revised.
2.1.2. Trace elements
In this sub-section of trace elements, only Zinc and Selenium are presented, but what do the authors think about the relevance of Iron and Calcium, for example, in seminal composition?
R: The sub-section of trace elements has been revised and rewritten. A more expanded description of the importance of the trace elements was carried out.
Line 144-145: Which reference do the authors think may be relevant to support this statement?
R: A reference has been added to support this statement in the 2.2.2 section.
Lines 148-156: On these lines, please indicate the studies showing these results; name, year and type of study. In addition, I suggest that the authors rewrite this section for a better understanding of what they want to explain about Zinc.
R: A better description of references has been added as the reviewer suggested. The sub-section of trace elements has been revised and rewritten. A more expanded description of the importance of the trace elements was carried out.
Lines 160-162: Which studies? No studies are presented in the text. Please expand on this information.
R: The studies were referenced accordingly.
2.1.3. Other BFCs
Lines 164-171: In this paragraph, please present the studies as suggested in the previous sections.
R: The author name of the study was added to unify results.
Lines 172-174: Add references in the presentation to Coenzyme Q10 and those that indicate the synthesis of Coenzyme Q10 and the foods where it is abundant, as explained in the manuscript.
R: The References have been added.
Lines 174-180: The presentation of these studies with Coenzyme Q10 should be unified. I suggest the authors rewrite this paragraph.
R: The paragraph has been rewritten.
Lines 182-185: I need references to be added in these two sentences to indicate the presence of PUFAs, as well as their relevant role in the sperm membrane.
R: The references have been added.
Lines 186-191: The authors present only one study in this review in relation to sperm quality and low omega-3 concentrations. Are they confident that they have conducted a commensurate search to present these results? I suggest reviewing PMID: 21219381, PMID: 25405715, PMID: 22416013 and PMID: 27792396.
R: The omega-3 discussion has been reviewed and the studies PMID: 21219381, PMID: 25405715, PMID: 22416013 and PMID: 27792396 added.
Lines 192-194: Please support these two sentences with relevant references.
R: The references have been added.
Lines 197-203: I suggest that you present the Balercia and Lenzi studies in a different way, organising the information they indicate.
R: The Balercia and Lenzi studies were rewritten.
Lines 212-221: The authors discuss the meta-analyses of Buhling and Smits, but only present the results found in Buhling's meta-analysis. What results are found in Smits' meta-analysis? Similar to Buhling's? Different? Explain them and add to the meaning of this paragraph.
R: The discussion of Smits revision has been added.
“The revision of Smits et. al. (2019) concluded that antioxidant supplementation taken by subfertile man may increase the rates of pregnancy, however, the evidence are low-quality and small clinical trials [63].”
- Diet and male reproductive epigenetics
3.1. Sperm-specific epigenetics
Line 229: The first time RNA appears in the text, explain its meaning as ribonucleic acid (RNA).
R:
Line 235: "Intrauterine" in "male gametogenesis" is a term that is not in line with the objective of the review presented by the authors. I suggest changing or modifying the meaning of the paragraph where this term appears.
R: We have removed information on intrauterine stage and focused on spermatogenesis.
Lines 242-244: The authors present just two studies showing the advancement of science and epigenetic information associated with spermatogenesis. I suggest to expand this information with other relevant studies; PMID: 33835194, PMID: 24908726, PMID: 29717022
R: Thanks for the suggestion. The references have been inserted.
Lines 248-251: "During the intrauterine period?" I suggest the authors modify the meaning of this sentence, as the manuscript should focus on spermatogenesis.
R: We have removed information on intrauterine stage and focused on spermatogenesis.
Lines 251-261: In this sequence of sentences, I do not perceive that an extensive literature search has been carried out to justify the use of these references presented.
R: We have updated the references in this key part of the manuscript.
Line 259: Please use the meaning of HPG the first time it is used in the manuscript.
R: Thank you for this remark. The modification was made.
Lines 262-269: I suggest that the authors reposition the quotations used for these phrases [8,55,57] in such a way that it is understood for which phrase is the most appropriate, as the way they are presented seems confusing for the reader.
R: Thank you for the suggestion. The references have been reorganized in the text.
Line 270: What does ncRNA mean? Show the meaning of non-coding RNA the first time it is used in the manuscript as non-coding RNA (ncRNA). Where is the supporting reference for this sentence?
R: Thank you for the suggestion. The modifications were made and the reference was placed in the text.
Line 275: I suggest the authors to modify or change the position of figure 1 at the end of this paragraph because it has a better interpretation in this paragraph.
R: Excellent suggestion. The change was made.
3.2. Micronutrients and BFCs epigenetic modulation in male germ cells
Line 283: The first sentence is relevant and important enough to support it with a reference. Consider PMID: 16251634
R: Thank you for the suggestion. The insertion of the reference has been made.
Why, if folate is so important in epigenetic modulation related to male gametogenesis, is it not explained in the previous subsection 2.1. Effects of BFCs on male gametogenesis? Justify this aspect and modify.
Line 291: What study? what author? It is relevant to show this study carried out in mice that serves to relate it to spermiogenesis in humans based on epigenetics and to show a unified presentation of the works with the authors throughout the manuscript.
R: I appreciate the comment. Modifications in the paragraph were made also taking into consideration the demands of reviewer 1.
- Paternal interventions with BFCs as a potential epigenetic strategy to improve health and prevent chronic disease in the offspring
4.1. Nutrition and Paternal Origins of Health and Disease (PoHAD)
Lines 302-304: The authors present only one reference that supports this statement about the DOHaD paradigm?
R: Thank you for the suggestion. The insertion of the reference has been made.
Lines 318-318: What several studies do the authors suggest in this sentence? The authors only present the Yeshurun study from 2017, which is a mouse study. I suggest authors modify this sentence or search for references that are relevant to this topic. Same suggestion as presented in line 319.
R: Thank you for the suggestion. The insertion of new references has been made.
Lines 320-324: I suggest rewriting this paragraph by adding relevant references, as suggested above.
R: Robust references have been added that support what is written in this paragraph.
Lines 325-327: I suggest you delete this sentence, because it is not relevant to present the following sections at this point.
R: The sentence was deleted.
Delete the spaces between section 4.1 and 4.2 (lines 328-330).
R: The modification was made.
4.2. Malnutrition and PoHAD
If this subsection only presents studies in mice, it should be rewritten to show the relevance and importance of conducting studies in humans.
R: The majority of studies in this context were performed in rodents. However we mention the The study “Newborn Epigenetic Study (NEST) cohort” [77,78] that was performed in humans and now better contextualize this lack of human studies and the need for the field of POHaD to expand clinical studies to better understand environmental factors including diet on male reproductive physiology and descendants health.
Lines 332-333: Which majority? The authors keep referencing Yeshurun 2017 study which is not a systematic review and was carried out on mice. Please rewrite this presentation.
R: Robust references have been added that support what is written in this sentence.
Lines 334-339: Shouldn't the authors focus more on alterations in sperm miR profiles rather than alterations in mammary gland morphology? Are there no human studies? I suggest the authors carry out a bibliographic search on this topic to retrieve relevant studies in humans and rewrite these lines.
R: In the article the altered microRNAS in the sperm of the LP group is not named. What is reported is the amount of miRnas that were differentially expressed. This information has been added in the manuscript. We chose not to remove the information about the female offspring since the topic deals with paternal programming.
Human studies are scarce, but just below in the text are two of the main human studies - “Newborn Epigenetic Study (NEST) cohort” [77,78].
Line 341: What does ACE and RAS pathway mean? Name the meaning of these acronyms the first time they appear in the manuscript.
R: Angiotensin-converting enzyme - ACE and renin–angiotensin system (RAS). Were placed in the text.
Lines 353-362: Studies carried out in humans. Perfect. You should consider separating between human and mouse studies if you decide to keep this section for better understanding.
R: Excellent suggestion. The human study was put separately in the last paragraph of the subtopic.
Lines 363-367: The authors join the above studies with the study by Wu et al. which is done on mice. I suggest rewriting this section 4.2 so as not to conflate the results found in mice and human studies.
R: The human and animal studies have already been properly separated. Thank you.
Lines 369-374: A 2018 study is not a recent study. In addition, it is another work presented in mice.
R: We apologize. The correction was made in the text.
Lines 375-379: The research group's study is also carried out on mice. How could it be extrapolated to future human studies? Justify these suggestions in this section.
We have now discussed how studies in humans could be conducted based on our own study. Please see page 20 (lines 428-431).
4.3. Micronutrients and Bioactive Food Compounds and PoHAD
I recommend the authors in this subsection to present the limitations of the scarcity of human studies and future lines of research to overcome them.
R: Indeed, studies showing the action of BFCs on paternal programming are scarce. Thank you for your comment. We highlighted this point at the end of the subtopic and in the conclusion. “Clinical and in vivo studies focusing on BFC's and paternal programming are still scarce. We highlight that dietary supplementation with BFC's and micronutrients in expecting fathers requires more in-depth studies to elucidate the underlying mechanisms that may alter the sperm epigenome and the impact the programming of future generations.”
Lines 386-400: The authors present only one study in rats. Do you think that with only one study it is relevant to present data? I suggest that they add more information on macronutrients and BFCs as suggested in previous sections and their relationship to epigenetics and spermatogenesis. The data presented by the authors are insufficient.
R: Even in vivo studies are scarce in the new field of PoHAD, so we found it necessary to put them in even though they are few.
Lines 401-411: This study carried out by McPherson et al. It has also been done in mice. Consider presenting human work or showing limitations to the scarcity of human work in relation to the micronutrients and BFCs presented in this section for the future.
R: We highlighted the scarcity of the studies at the end of the subtopic and in the conclusion.
Lines 413-425: Similar to the above considerations.
- Conclusions
The authors should separate the key points of the conclusions in different paragraphs for the different results presented.
The limitations that exist in the review presented by the authors in human studies that show the relationship between epigenetics and spermatogenesis and infertility problems and how they could be improved with food should be present. I believe that these aspects remain to be demonstrated
R: We have separated the conclusions accordingly and discuss the need for human studies that show the relationship between epigenetics and spermatogenesis and infertility problems and how they could be improved with food in the paragraph before the Conclusion.
Round 2
Reviewer 1 Report
Thank you for addressing most of my comments. I have a small number of remaining comments for you to address.
General comments
The manuscript still contains grammatical errors throughout e.g. line 285 “the revision” should say review. Also, there are errors in using the words male, man, men etc. throughout the text. Please proofread for grammar and language
Please be more consistent with acronyms, e.g. you use ‘BFCs’ and then ‘bioactive food compounds’ and then ‘BFCs’ again.
Please show our comments on Figure 1. It is currently missing.
Please address my comment related to Figure 1 “you mention phytochemicals for the first time here. You should therefore mention it in the paper as well” The answer you provided does not match my comment.
Specific comments
Line 416 and 417: please correct the mouse gene nomenclature
Line 462 “highly supplemented” should be changed to ‘supplemented in excess’
Line 464: I do not understand your reference to a ‘U-shaped pattern’ – please explain in more detail.
Line 472: please replace ‘Se’ with ‘selenium’.
Author Response
We would like to thank again very much Reviewer 1 for the very important comments and recommendations that improved the quality of our manuscript. Please find below our responses to all your questions.
General comments
The manuscript still contains grammatical errors throughout e.g. line 285 “the revision” should say review. Also, there are errors in using the words male, man, men etc. throughout the text. Please proofread for grammar and language
R: All corrections have been now made. Thank you.
Please be more consistent with acronyms, e.g. you use ‘BFCs’ and then ‘bioactive food compounds’ and then ‘BFCs’ again.
R: Thank you for the observation. Acronyms have been standardized throughout the text.
Please show our comments on Figure 1. It is currently missing.
R: We apologize. Something must have happened for the answers referring to figure 1 not have been contemplated in our previous response letter. The comments regarding figure 1 with their respective answers are as follows:
Figure 1.
- you mention phytochemicals for the first time here. You should therefore mention it in the paper as well
R: The new sentence is “BFCs play a role in male fertility and fetal programming through epigenetic mechanisms with alterations in DNA methylation, histone modifications, and microRNAs and is capable to restore metabolic health disturbances in offspring induced by stressors during early life”. We have now unified all dietary factors under the term bioactive food compounds and removed the term phytochemicals form the figure 1, as it was never mentioned in the text.
- the figure has grammatical errors, can the authors please revise grammar/spelling
R: Thank you for the observation. All errors have been corrected.
- gene symbols should be italicised and capitalised if talking about humans, which the diagram appears to be
R: Thank you. The correction was made.
- formatting issues with spaces between words
R: Thank you. The correction was made.
Please address my comment related to Figure 1 “you mention phytochemicals for the first time here. You should therefore mention it in the paper as well” The answer you provided does not match my comment.
R: We have chosen to use only the term bioactive food compounds since phytochemicals fall under the definition of BFC "bioactive food compounds (BFCs) that include micronutrients and non-nutrients have been highlighted as a potential strategy to protect against oxidative and inflammatory damage(...)” – lines 26-27, which in this case would be BFC non-nutrients. For this reason, the word "phytochemicals" has been removed from the figure.
Specific comments
Line 416 and 417: please correct the mouse gene nomenclature
R: Thank you for this observation. The correction has been made.
Line 462 “highly supplemented” should be changed to ‘supplemented in excess’
R: Thank you for this remark. The correction has been made.
Line 464: I do not understand your reference to a ‘U-shaped pattern’ – please explain in more detail.
R: The sentence was rewritten and became “This unbalance is a key aspect when considering supplementing future fathers with micronutrients, as both dietary deficiencies and excess may lead to some deleterious outcomes.” In any event, this U-shapped pattern happens when both deficiency or excess of a specific micronutrient lead to the same outcome.
Line 472: please replace ‘Se’ with ‘selenium’.
R: Thank you for this observation. The correction has been made.
Reviewer 2 Report
The authors have responded correctly to the reviewer's suggested revisions.
Congratulations on the improvement of the manuscript.
Author Response
We would like to thank Reviewer 2 for the important comment.